# Potential of MALDI-TOF MS biotyping to detect deltamethrin resistance in the dengue vector *Aedes aegypti*

**Lionel Almeras**[1,2,3]*, **Monique Melo Costa**[1,2,3], **Rémy Amalvict**[1,2,3,4], **Joseph Guilliet**[5], **Isabelle Dusfour**[6], **Jean-Philippe David**[5], **Vincent Corbel**[7,8]

1 Département Microbiologie et Maladies Infectieuses, Institut de Recherche Biomédicale des Armées, Unité Parasitologie et Entomologie, Marseille, 13005, France, 2 Aix Marseille University, IRD, AP-HM, SSA, VITROME, Marseille, 13005, France, 3 IHU-Méditerranée Infection, Marseille, 13005, France, 4 Centre National de Référence du Paludisme, Marseille, 13005, France, 5 Laboratoire d'Ecologie Alpine, UMR UGA-USMB-CNRS 5553, Université Grenoble Alpes, Grenoble, 38041, France, 6 Institut Pasteur de la Guyane, Vectopôle Amazonien Emile Abonnenc, Unité de Contrôle et Adaptation des Vecteurs, Cayenne, France, 7 MIVEGEC, IRD, CNRS, University of Montpellier, Montpellier, France, 8 Laboratório de Fisiologia e Controle de Artrópodes Vetores (Laficave), Instituto Oswaldo Cruz (IOC), Fundacao Oswaldo Cruz (FIOCRUZ), Avenida Brasil, Rio de Janeiro–RJ, Brazil

* almeras.lionel@gmail.com

**Data Availability Statement:** All relevant data are within the manuscript and its Supporting Information files.

## Abstract

Insecticide resistance in mosquitoes is spreading worldwide and represents a growing threat to vector control. Insecticide resistance is caused by different mechanisms including higher metabolic detoxication, target-site modification, reduced penetration and behavioral changes that are not easily detectable with simple diagnostic methods. Indeed, most molecular resistance diagnostic tools are costly and labor intensive and then difficult to use for routine monitoring of insecticide resistance. The present study aims to determine whether mosquito susceptibility status against the pyrethroid insecticides (mostly used for mosquito control) could be established by the protein signatures of legs and/or thoraxes submitted to MALDI-TOF Mass Spectrometry (MS). The quality of MS spectra for both body parts was controlled to avoid any bias due to unconformity protein profiling. The comparison of MS profiles from three inbreeds *Ae. aegypti* lines from French Guiana (IRF, IR03, IR13), with distinct deltamethrin resistance genotype / phenotype and the susceptible reference laboratory line BORA (French Polynesia), showed different protein signatures. On both body parts, the analysis of whole protein profiles revealed a singularity of BORA line compared to the three inbreeding lines from French Guiana origin, suggesting that the first criteria of differentiation is the geographical origin and/or the breeding history rather than the insecticide susceptibility profile. However, a deeper analysis of the protein profiles allowed to identify 10 and 11 discriminating peaks from leg and thorax spectra, respectively. Among them, a specific peak around 4870 Da was detected in legs and thoraxes of pyrethroid resistant lines compared to the susceptible counterparts hence suggesting that MS profiling may be promising to rapidly distinguish resistant and susceptible phenotypes. Further work is needed to confirm the nature of this peak as a deltamethrin resistant marker and to validate the routine use of MS profiling to track insecticide resistance in *Ae. aegypti* field populations.

**Funding:** AL received the following award This work has been supported by the Délégation Générale pour l'Armement (DGA), MSProfileR project, Grant no PDH-2-NRBC-2-B-2201 This work was also supported by the WIN (Worldwide Insecticide resistance Network).

**Competing interests:** The authors have declared that no competing interests exist.

**Abbreviations:** *Ae*, *Aedes*; CCI, Composite Correlation Index; CV, cross-validation; DDT, dichlorodiphenyltrichloroethane; GA, genetic algorithm; IR, insecticide resistance; kdr, knockdown resistance; LSV, Log Score Value; MALDI-TOF MS, Matrix Assisted Laser Desorption/ Ionization Time-of-Flight Mass Spectrometry; MSP, Main Spectra Projection; PCA, principal component analysis; RC, recognition capability; RSDB, reference spectra database; VGSC, voltage-gated sodium channel.

## Introduction

*Aedes (Ae.) aegypti (Diptera: Culicidae)* is an urban mosquito species that can transmit viruses to humans causing infectious diseases, such as dengue, yellow fever, Zika, and chikungunya. The geographic distribution of this pest is the widest ever recorded in history and it represents an increasing public health threat [1]. The absence of specific antiviral treatments and the lack of vaccines or when available the low vaccination coverage underlined that the best protection method remains to avoid human exposure to *Ae. aegypti* bites [2]. Larval source management, community mobilization and chemical insecticides remain the first line of defense against this pest [3,4]. Unfortunately, the continuous use of public health pesticides for more than 40 years increased aversion of citizens to strategies based solely on insecticides because of their potential impacts on the environment and global health [4]. Furthermore, the use of the same insecticides in vector control for decades has selected mosquito resistances to all public health insecticides. Resistance in *Ae. aegypti* and *Ae. albopictus* is now present in at least 57 countries in South East Asia, Africa, the Americas and the Caribbean, where the burden of arboviral diseases is the highest [5]. Evidence of reduced susceptibility to insecticides has also been recently reported in invasive *Aedes* mosquitoes in Europe, especially from Italy, Greece and Spain [6,7], hence confirming that resistance is spreading rapidly across continents. In this context, there is a need for more adequate, scalable and affordable tools for tracking insecticide resistant mosquitoes in the field to prevent further spread.

Different mechanisms are known to confer resistance to chemical insecticides. One of the most widespread and known mechanisms is knockdown resistance (*kdr*) causing resistance to dichlorodiphenyltrichloroethane (DDT) and pyrethroids [8]. The mechanism is associated with point mutations affecting the gene encoding the voltage-gated sodium channel (VGSC), which is involved in the beginning and propagation of action potentials in the nervous system [8]. The mechanism was originally discovered in the housefly and then identified in a large number of arthropods including mosquitoes [9,10]. In *Ae. Aegypti*, several *kdr* mutations are known to confer resistance to pyrethroids and DDT including the V410L(a substitution of a valine to leucine at position 410), S989P, V1016I/G (i.e. a substitution of a valine to either iso-leucine or glycine at position 1016) and F1534C (i.e. a substitution of a phenylalanine to cysteine at position 1534) mutations that were found in different regions of the world [11]. In addition, metabolic resistance through the over-expression of detoxifying enzymes belonging to Monooxygenases (P450s), Glutathione-S-Transferases (GSTs) and Carboxylesterases (CCEAs) can also confer high level of resistance to various classes of insecticides including pyrethroids [12,13]. Recent studies showed that additional mechanisms such as reduced insecticide penetration due to change in the thickness and/or composition of the cuticle [14,15] confer the insect the capacity to resist to multiple classes of insecticides.

Current methods for resistance monitoring rely on biological, biochemical and molecular assays that all have technical and/or operational constraints (*e.g.* lack of sensitivity or specificity, cost, low throughput). The strength and weakness of each method were previously reported by Dusfour et al [16]. Developing novel affordable and accurate strategies to detect resistant mosquitoes at high-throughput would facilitate the implementation of timely and locally adapted insecticide resistance management strategies.

Recently, an innovative method based on the analysis of protein profiles obtained by Matrix-Assisted Laser Desorption/Ionization time-of-flight mass spectrometry (MALDI-TOF MS) profiling, was applied to arthropod identification [17,18]. Since 2013, we conducted pioneering studies, applying successfully this approach to the identification of several arthropod family such as mosquitoes, ticks, fleas or culicoides [19–22] as well as to the identification of blood source of engorged mosquitoes [23]. The principle of the classification is based on

matching of query MS profiles with a MS reference spectra database (RSDB). The correct classification requires however that MS spectra are intra-species reproducible and inter-species specific. As for most phenotypic approaches, MS protein profiles could vary according to body part, developmental stages or sample preparation mode [24]. Then, to compare and to share MS results, a standardization of the protocols were previously established for mosquitoes [25,26]. We also showed that the independent submission of legs and thorax from the same specimen to MALDI-TOF MS [27] can improve the identification rate and confidence level, that may be decisive for discriminating similar phenotypes, such as cryptic species [26,28].

In this context, the aim of the present work was to determine whether resistance to the pyrethroid ester insecticide, could be detected in *Ae. aegypti* by analyzing the protein signatures of legs and/or thoraxes resulting from MALDI-TOF MS. In this way, MS profiles from four *Ae. aegypti* colonies, including one susceptible reference laboratory line from French Polynesia and three inbreeding lines from French Guiana showing distinct resistance phenotypes to deltamethrin were compared.

## Material and methods

### Mosquito laboratory breeding

Four Ae. aegypti colonies, including the susceptible reference laboratory line Bora-Bora (BORA) and three isofemale lines from French Guiana with distinct deltamethrin resistance phenotypes were used. The first two Ae. aegypti lines, (IR13 and IR03 lines) were obtained from gravid females collected in the Ile Royale, an island off the coast of Cayenne, by the Pasteur Institute of French Guiana [29]. According to WHO manual for monitoring insecticide resistance, the IR13 line presented a slight tolerance to deltamethrin (mortality to discriminating concentration of deltamethrin $\geq$ 90%, but lower than 98%) and was considered as susceptible, whereas the IR03 line was considered as resistant to deltamethrin (mortality <90% to the discriminating concentration of deltamethrin) [30]. Previous study supported the absence of the V410L, S989P and V1016I/G kdr mutations in both IR03 and IR13 lines. However, both IR13 and IR03 lines initially carried the F1534C mutation at moderate frequency [29]. Another resistant line deprived from the F1534C kdr mutation (IRF) was created by internal crossing of the IR03 line. Despite the absence of the F1534C mutation, the IRF line remained resistant to deltamethrin [31], supporting the importance of metabolic resistance alleles in both resistant lines as previously shown [12,29,32,33]. All mosquito lines were reared using standard methods [27]. For eggs production, blood-meals were given through a Parafilm-membrane (hemotek membrane feeding systems, Discovery Workshops, UK) as previously described [24]. Larvae were reared until the pupal stage in trays containing 1liter distilled water supplemented with fish food (TetraMinBaby, Tetra Gmbh, Herrenteich, Germany). Pupae were daily collected and transferred to mosquito cages (Bug Dorm 1, Bioquip products). For mass spectrometry (MS) analysis, pupae female, distinguished by sexual dimorphism, were transferred to mosquito cages. Twenty virgin, non-blood fed, 3 days-old females per line were collected and frozen at -20°C until MS analysis.

### Mosquito dissection

Legs and thoraxes of each mosquito were processed as previously described [34]. Briefly, Aedes specimens were individually dissected, under a binocular loupe, with a sterile surgical blade. For each specimen, legs and thorax (without wings) were transferred in distinct 1.5 mL Eppendorf tubes for MALDI-TOF MS analysis. The remaining body parts (abdomens, wings and heads) were preserved for molecular analyses.

## Knockdown resistance (kdr) genotyping

Genomic DNA was extracted individually from the remaining body parts (abdomen, wings and head) of 20 individual adult specimens per line using the QIAamp DNA tissue extraction kit (Qiagen, Hilden, Germany), according to the manufacturer's instructions. The kdr genotyping of V410L (a substitution of a valine to leucine at position 410), V1016I/G (i.e. a substitution of a valine to either isoleucine or glycine at position 1016) and F1534C (i.e. a substitution of a phenylalanine to cysteine at position 1534) were conducted by standard PCR (S1 Table) followed by sequencing as previously described [35].

## Sample homogenization and MALDI-TOF MS analysis

Each body part (legs and thorax) was homogenized individually for 3 x 1 minute at 30 Hertz using TissueLyser (Qiagen) and glass beads (#11079110, BioSpec Products, Bartlesville, OK, US) in a homogenization buffer composed of a mix (50/50) of 70% (v/v) formic acid (Sigma) and 50% (v/v) acetonitrile (Fluka, Buchs, Switzerland) according to the standardized automated setting as described previously [27]. After sample homogenization, a quick spin centrifugation at 200 g for 1 min was then performed and 1 μL of the supernatant of each sample was spotted on the MALDI-TOF steel target plate in duplicate (Bruker Daltonics, Wissembourg, France). After air-drying, 1 μL of matrix solution composed of saturated α-cyano-4-hydroxycinnamic acid (Sigma, Lyon, France), 50% (v/v) acetonitrile, 2.5% (v/v) trifluoroacetic acid (Aldrich, Dorset, UK) and HPLC-grade water was added. To control matrix quality (i.e. absence of MS peaks due to matrix buffer impurities) and MALDI-TOF apparatus performance, matrix solution was loaded in duplicate onto each MALDI-TOF plate alone. Protein mass profiles were obtained using a Microflex LT MALDI-TOF Mass Spectrometer (Bruker Daltonics, Germany), with detection in the linear positive-ion mode at a laser frequency of 50 Hz within a mass range of 2–20 kDa. The setting parameters of the MALDI-TOF MS apparatus were identical to those previously used [36]. Daily, an automatic calibration of MALDI-TOF apparatus was done using a Bruker Bacterial Test Standard (BTS, ref: #8255343). In addition, to control quality of mosquito homogenization step and MALDI-TOF apparatus performance of each plate, highly relevant and correct identification score (LSVs >2.0) should be obtained for at least more than half of the legs or thoraxes from Ae. aegypti BORA specimens queried against the reference MS database. These identification parameters were used as internal QC to validate each plate.

## MS spectra analysis

MS spectra profiles were firstly controlled visually with flexAnalysis v3.3 software (Bruker Daltonics). MS spectra were then exported to ClinProTools v2.2 and MALDI-Biotyper v3.0. (Bruker Daltonics) for data processing (smoothing, baseline subtraction, peak picking). MS spectra reproducibility was assessed by the comparison of the average spectral profiles (MSP, Main Spectrum Profile) obtained from the two spots of each specimen from legs and thorax according to Aedes lines with MALDI-Biotyper v3.0 software (Bruker Daltonics). MS spectra reproducibility and specificity were achieved using cluster analyses and Composite Correlation Index (CCI) tool. Cluster analyses (MSP dendrogram) were performed based on comparison of the MSP given by MALDI-Biotyper v3.0. software and clustered according to protein mass profile (i.e. their mass signals and intensities). The CCI tool from MALDI-Biotyper v3.0. software was also used, to assess the spectral variations within and between each sample group, as previously described [20,27]. CCI matrix was calculated using MALDI-Biotyper v3.0. software with default settings (mass range 3.0–12.0 kDa; resolution 4; 8 intervals; auto-correction off). Higher correlation values (expressed by mean ± standard deviation – SD) reflecting higher

reproducibility for the MS spectra, were used to estimate MS spectra distance between lines. To visualize MS spectra distribution according to insecticide susceptibility, a principal component analysis (PCA) from ClinProTools v2.2 software was used with the default settings.

The spectra were then analysed with the genetic algorithm (GA) model, which displayed a list of discriminating peaks between the two insecticide susceptible lines (BORA and IR13) and the two resistant lines (IR03 and IFR). A manual inspection and validation of the selected peaks by the operator gave a recognition capability (RC) value together with the highest cross-validation (CV) value. The presence or absence of all discriminating peak masses generated by the GA model was controlled by comparing the average spectra from line per body-part.

## Database creation and blind tests

The reference MS spectra were created using spectra from two specimens per line and per body parts (legs and thorax) using MALDI-Biotyper software v3.0. (Bruker Daltonics). MS spectra were created with an unbiased algorithm using information on the peak position, intensity and frequency. The remaining MS spectra per line and body part were queried against these reference MS spectra and a classification as deltamethrin-resistant or–susceptible was done according to the result of spectral matching with DB. The reliability of sample classification was determined using the Log Score Values (LSVs) given by the MALDI-Biotyper software v.3.0, corresponding to a matched degree of mass spectra between the query and the reference spectra from the DB. LSVs ranging from 0 to 3 were obtained for each spectrum of the samples tested. According to previous studies [28,37], LSVs greater than 1.8 were considered reliable for species identification. For evaluating it performance, the sensitivity, the specificity, the accuracy and the Cohen's κ coefficient, corresponding to the degree of agreement [38], were calculated.

## Statistical analyses

After verifying that the LSVs in each line did not follow a Gaussian distribution (Shapiro-Wilk test), Wilcoxon matched-pairs signed-rank tests were computed when appropriate using GraphPad Prism v7.00 (GraphPad Software, La Jolla California USA,). Frequencies were compared by the Chi-square test. All differences were considered significant at $p < 0.05$. For detection of discriminant MS peaks, statistical tests from ClinProTools v2.2 software, including t-test (ANOVA), the Wilcoxon or Kruskal-Wallis (W/KW) test and the Anderson-Darling (AD) test were applied, to short peaks among profiles. To consider a peak as discriminant, it should obtain a significant p-value ($<0.05$) in the AD test but also in the W/KW or ANOVA tests [39]. Among these discriminant MS peaks were selected those which presented a fold change upper than 1.3-fold between susceptible and resistant lines [40].

## Ethical approval

The present work included only four colonies of Aedes aegypti mosquitoes, which were obtained from laboratory rearing. These mosquito species were not classified as endangered or protected species.

## Results

### Low MS spectra diversity between *Ae. aegypti* lines

Legs and thoraxes from 20 specimens per line (BORA, IR13, IR03 and IRF) were submitted independently to MALDI-TOF MS analysis (Fig 1). At the exception of the legs from one specimen from the IR03 line, MS profiles of high intensity ($>2000$ a.u.) were obtained for all

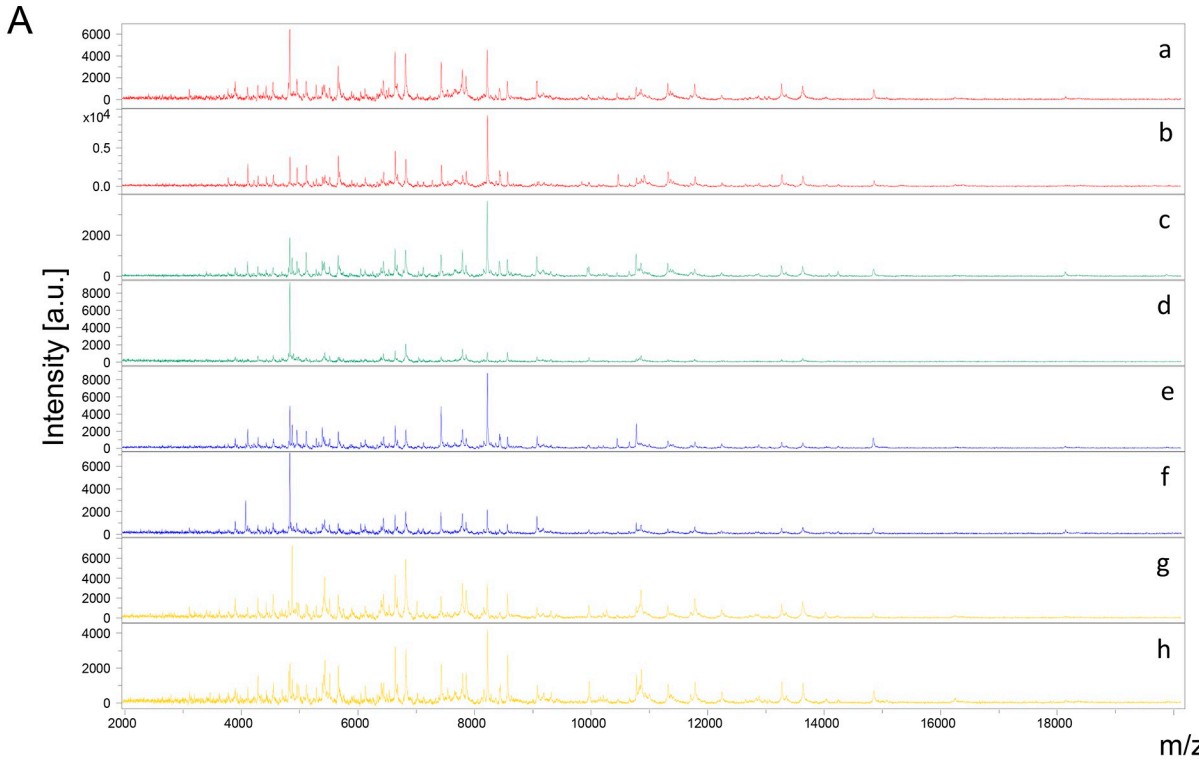

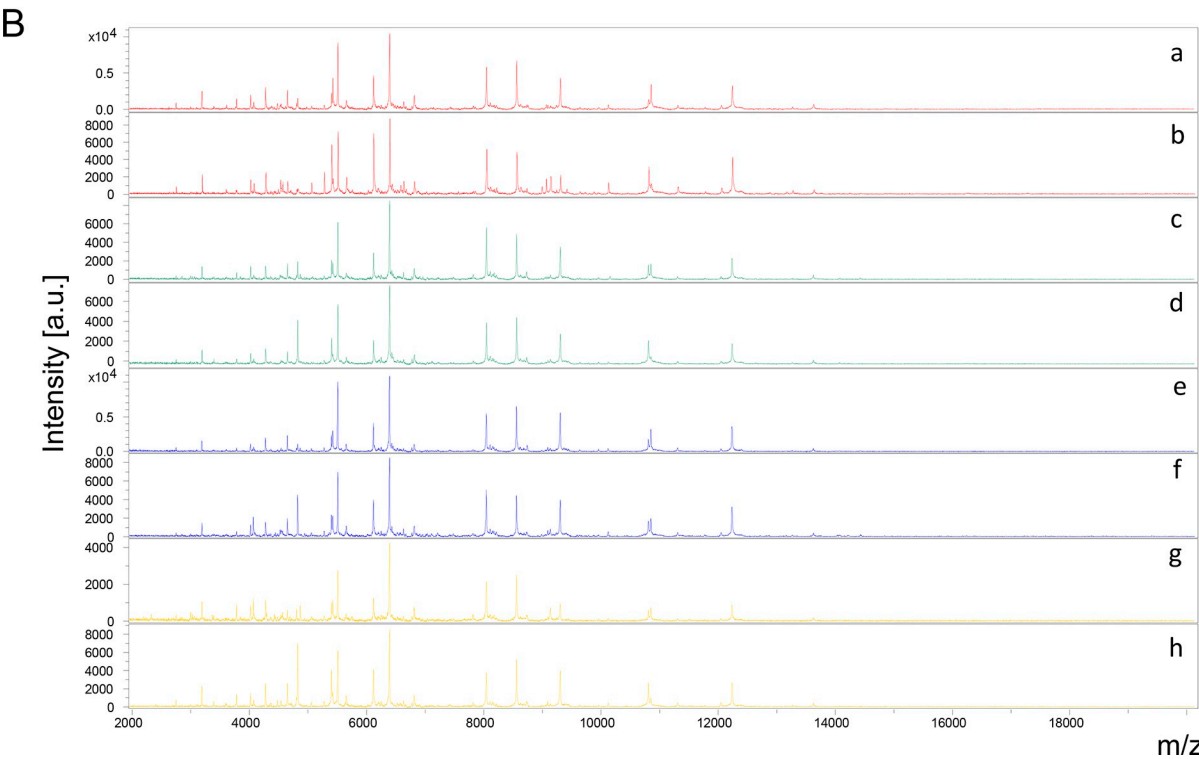

**Fig 1. Comparison of MALDI-TOF MS spectra from legs (A) and thoraxes (B) of four Aedes aegypti lines.** Representative MS spectra of two Ae. aegypti specimens per line, susceptible (BORA (a, b) and IR13 (c, d)) or resistant (IR03 (e, f) and IRF (g, h)) to deltamethrin. a.u., arbitrary units; m/z, mass-to-charge ratio.

samples from both body parts. To control MS spectra "quality", they were queried against our MS spectra database (DB) which includes reference MS spectra of legs and thoraxes from 16 distinct mosquito species [27,28] and notably Ae. aegypti. One hundred percent of the legs and thoraxes MS spectra matched with reference MS spectra from Ae. aegypti with respective body parts. Respectively, 97.5% (78/80) and 100% (80/80) of the MS spectra from legs and thoraxes reached the threshold LSV of 1.8 considered as a successful identification (Fig 2A and 2B) [21,25]. It is interesting to note that LSVs from thoraxes were significantly higher than those obtained for legs (Wilcoxon test, p<0.0001). More than 98% (79/80) of the thoraxes presented a LSV upper than 2.0 whereas only 71.25% (57/80) of the MS spectra from legs reached this last threshold.

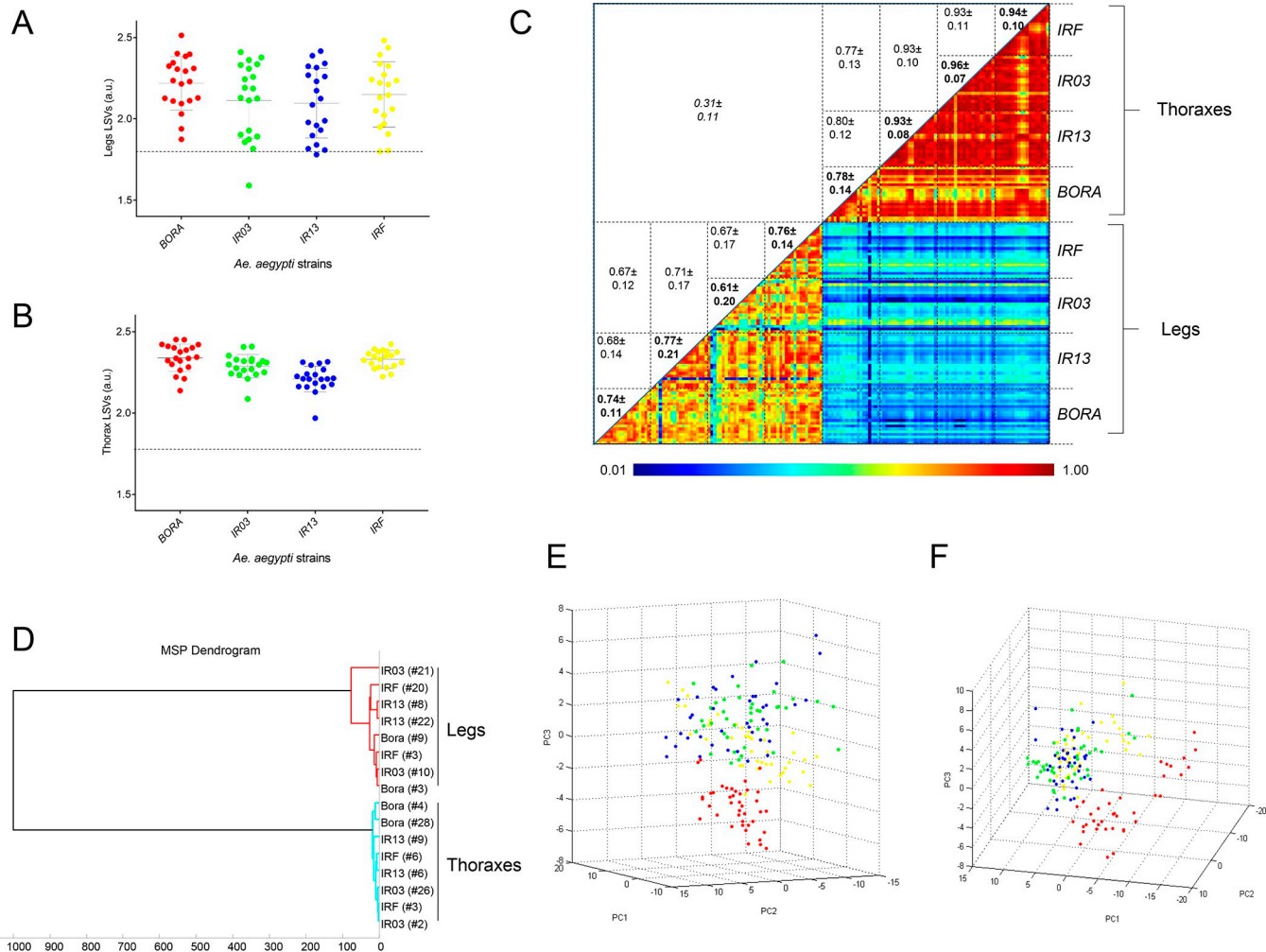

**Fig 2. Reproducibility and specificity of MALDI-TOF MS spectra from *Aedes aegypti* lines according to body part.** LSVs obtained following homemade MS reference database query with MS spectra of the four Ae. aegypti lines from legs (A) and thoraxes (B). Twenty specimens per line were tested. Horizontal dashed lines represent the threshold value for reliable identification (LSV>1.8). LSVs, log score values; a.u., arbitrary units. (C) The same 20 MS spectra per line and body part were analysed using the composite correlation index (CCI) tool. Levels of MS spectra reproducibility are indicated in red and blue revealing relatedness and incongruence between spectra, respectively. The values correspond to the mean coefficient correlation and respective standard deviations obtained for paired condition comparisons. CCI were expressed as mean ± standard deviation. (D) MSP dendrogram of MALDI-TOF MS spectra from legs (red) and thoraxes (blue) of the four Ae. aegypti lines. Two specimens per lines and per body part are presented. The distance units correspond to the relative similarity of MS spectra. The dendrogram was created by Biotyper v3.0 software. Principal Component Analysis (PCA) dimensional image from thoraxes (E, n = 20) and legs (F, n = 20) MS spectra of the four lines. Red, green, blue and yellow dots correspond to BORA, IR13, IR03 and IRF Ae. aegypti lines, respectively.

To assess MS spectra reproducibility according to each Ae. aegypti line, CCI matrix, MSP dendrogram and PCA were performed. The low CCI obtained for the comparisons of paired MS spectra between thoraxes and legs (mean ± SD: 0.31 ± 0.11) sustained the specificity of each body-part protein profiles. As expected, higher CCI were obtained for specimens from the same line than between lines for each body part, excepted for IR03 legs (mean CCI ±SD = 0.61±0.20, Fig 2C). This lower CCI value was attributed to the lower quality of MS spectra from one sample of the IR03 line. Moreover, values from thorax CCIs were more elevated than those of legs supporting that MS profiles from thoraxes were more reproducible. Interestingly, higher thorax CCIs were obtained between the two resistant lines (IR03 and IRF) and the IR13 susceptible line (mean ± SD: 0.93 ± 0.10), compared to the laboratory susceptible line (BORA) (mean ± SD: 0.77 ± 0.13). Similarly, the reproducibility of leg MS profiles was higher between lines originating from French Guiana.

To assess the reproducibility and specificity of the MS spectra from legs and thoraxes according to their deltamethrin resistance phenotype, a cluster analysis was performed. Two specimens per line were used for building a MSP dendrogram (Fig 2D). Legs and thoraxes clustered in distinct branches confirming the specificity of the MS spectra per body part. However, no gathering of the spectra was noticed according to the resistance status or line, excepted for BORA using the thoraxes. The PCAs performed per body part with spectra from all samples showed two clusters. One cluster encompassed the three lines from French Guiana (IR03, IR13 and IRF) and another one included the BORA line for spectra from thoraxes (Fig 2E) and legs (Fig 2F). These results highlighted that overall comparisons of MS profiles did not allow to clearly distinguish specimens according to their deltamethrin resistance status whatever the body-part tested.

## Identification of discriminant MS peaks between deltamethrin-resistant and -susceptible lines

To assess whether it was possible to identify discriminating MS peaks according to the deltamethrin resistance status, the MS spectra from the 20 specimens per line were analyzed for each body part (leg and thorax), using ClinProTools software. Then, the average spectrum from the insecticide-resistant lines (IR03 and IRF) were compared to the susceptible lines (BORA and IR13). A total of 99 and 118 peaks were detected in the average spectra of legs and thoraxes, respectively. MS peaks were considered as discriminant if they have a fold change upper than 1.3-fold in either direction between the two groups and if these variations were considered as statistically significant according to criteria defined previously (see material and methods). After verification of the peak report, 10 and 11 MS peaks from legs and thoraxes respectively (Tables 1 and 2) were considered of significant different intensity between the two groups. To assess whether these MS peaks could be discriminatory among these groups, they were included in the genetic algorithm (GA) model from ClinProTools 2.2 software. The combination of the presence/absence of these MS peaks from each body part lead to high RC and CV values for legs (90.0% and 98.1% respectively) and also for thoraxes (92.5% and 97.5%, respectively).

Interestingly, one potential discriminant MS peak was shared between the two body parts with a m/z of about 4870 Da, corresponding to peak #16 in legs (Fig 3A–3C) and peak #29 in thorax (Fig 3D–3F). As the start and end masses of the peak #16 (4871.1) and #29 (4869.5) were at m/z from 4859.2 to 4883.3 and from 4855.7 to 4883.0, respectively, it was considered that these two peaks are the same. The m/z of this discriminant peak found in both body parts was set at about 4870 Da. This 4870 Da peak was of greater intensity in the deltamethrin resistant lines than in the two susceptible lines for both body parts (Tables 1 and 2). Interestingly,

**Table 1. List of the discriminant MS peaks from legs between deltamethrin-resistant (IR03 and IRF) and–susceptible (BORA and IR13) *Ae. aegypti* lines.**

| Peak number | Mass (Da) | PTTA | PW/KW | PAD | Average peak intensity (mean ± SD in a.u.) | | Fold change |
|---|---|---|---|---|---|---|---|
| | | | | | R | S | Ratio R/S |
| 16 | 4871.1[#] | < 0.000001 | 0 | < 0.000001 | 11.1 ± 7 | 3.59 ± 1.94 | 3.09 |
| 22 | 5387.5 | 0.00517 | 0.0132 | < 0.000001 | 6.2 ± 3.24 | 4.76 ± 2.2 | 1.30 |
| 41 | 7019.9 | < 0.000001 | < 0.000001 | < 0.000001 | 2.97 ± 1.05 | 1.94 ± 0.57 | 1.53 |
| 59 | 9072. 5 | 0.000364 | 0.000132 | < 0.000001 | 5.51 ± 2.28 | 4.09 ± 2.1 | 1.35 |
| 64 | 9965.0 | 0.0000328 | 0.0000129 | 6.36E-06 | 3.09 ± 0.98 | 2.37 ± 0.89 | 1.30 |
| 70 | 10776.2 | 0.000345 | 0.00129 | < 0.000001 | 6.32 ± 4.17 | 3.99 ± 2.82 | 1.58 |
| 83 | 12242.5 | < 0.000001 | < 0.000001 | 0.000151 | 2.11 ± 0.58 | 1.53 ± 0.36 | 1.38 |
| 93 | 14036.2 | < 0.000001 | 0 | < 0.000001 | 1.42 ± 0.36 | 0.97 ± 0.18 | 1.46 |
| 96 | 14851.6 | 0.0000565 | 0.0000882 | < 0.000001 | 2.54 ± 1.17 | 1.78 ± 0.86 | 1.43 |
| 99 | 18145.0 | 5.79E-06 | 1.08E-06 | < 0.000001 | 1.27 ± 0.44 | 0.94 ± 0.34 | 1.35 |

#MS peaks for which mass-to-charge ratio (m/z) were similar with thoraxes MS peak list (see Table 2). Da, Dalton; PTTA, p-value obtained by t-test; PW/KW, p-value obtained by Wilcoxon/Kruskal-Wallis test; PAD, p-value obtained by Anderson-Darling test; a.u., arbitrary unit; R, deltamethrin-resistant lines; S, deltamethrin-susceptible lines.

this peak was among the MS peak presenting the higher fold change (>2.8 fold) between resistant and susceptible-lines for both body parts (Tables 1 and 2). Moreover, with a respective average peak intensity of 11.1 arbitrary units (a.u.) and 6.7 a.u. for the leg and thorax peaks at about m/z 4870, these peaks were classified in the top ten and top thirty for leg and thorax profiles from deltamethrin-resistant lines (IR03 ad IRF). Submitting this single peak to the GA model lead to RC and CV values of 78.2% and 85.5%, respectively for legs and 79.8% and 85.6%, respectively for thoraxes. The detection of this peak in the IR13 susceptible line though at a lower intensity confirmed that no single peak was found exclusive of resistant-group species but that the discrimination was more attributed to intensity variations.

**Table 2. List of the discriminant MS peaks from thoraxes between deltamethrin-resistant (IR03 and IRF) and–susceptible (BORA and IR13) *Ae. aegypti* lines.**

| Peak number | Mass (Da) | PTTA | PW/KW | PAD | Average peak intensity (mean ± SD in a.u.) | | Fold change |
|---|---|---|---|---|---|---|---|
| | | | | | R | S | Ratio R/S |
| 5 | 3026.9 | 0.000465 | 0.00208 | < 0.000001 | 1.96 ± 0.99 | 1.47 ± 0.49 | 1.33 |
| 16 | 4075.2 | 0.125 | 0.0112 | < 0.000001 | 6.73 ± 8.2 | 5.02 ± 2.38 | 1.34 |
| 19 | 4432.1 | 0.0415 | 0.0175 | < 0.000001 | 2.84 ± 2.03 | 2.15 ± 1.69 | 1.32 |
| 20 | 4446.3 | 0.0265 | 0.00243 | < 0.000001 | 3.01 ± 2.11 | 2.18 ± 2.03 | 1.38 |
| 24 | 4569.6 | 0.000179 | 0.0000745 | < 0.000001 | 4.37 ± 1.93 | 3.22 ± 1.5 | 1.36 |
| 29 | 4869.5[#] | < 0.000001 | 0 | < 0.000001 | 6.19 ± 4.22 | 2.19 ± 0.89 | 2.83 |
| 74 | 9066.7 | < 0.000001 | < 0.000001 | < 0.000001 | 1.7 ± 0.32 | 4.18 ± 2.92 | 0.41 |
| 75 | 9095.5 | < 0.000001 | < 0.000001 | < 0.000001 | 3.81 ± 1.74 | 2.28 ± 0.86 | 1.67 |
| 76 | 9140.8 | 9.98E-06 | < 0.000001 | < 0.000001 | 5.86 ± 3.84 | 3.38 ± 2.0 | 1.73 |
| 110 | 13267.5 | 2.19E-06 | 5.25E-06 | < 0.000001 | 1.13 ± 0.32 | 1.56 ± 0.62 | 0.72 |
| 114 | 14030.0 | < 0.000001 | < 0.000001 | 0.000017 | 0.89 ± 0.21 | 0.66 ± 0.11 | 1.35 |

#MS peaks for which mass-to-charge ratio (m/z) were similar with legs MS peak list (see Table 1). Da. Dalton; PTTA, p-value obtained by t-test; PW/KW, p-value obtained by Wilcoxon/Kruskal-Wallis test; PAD, p-value obtained by Anderson-Darling test; a.u., arbitrary unit; R, deltamethrin-resistant lines; S, deltamethrin-susceptible lines.

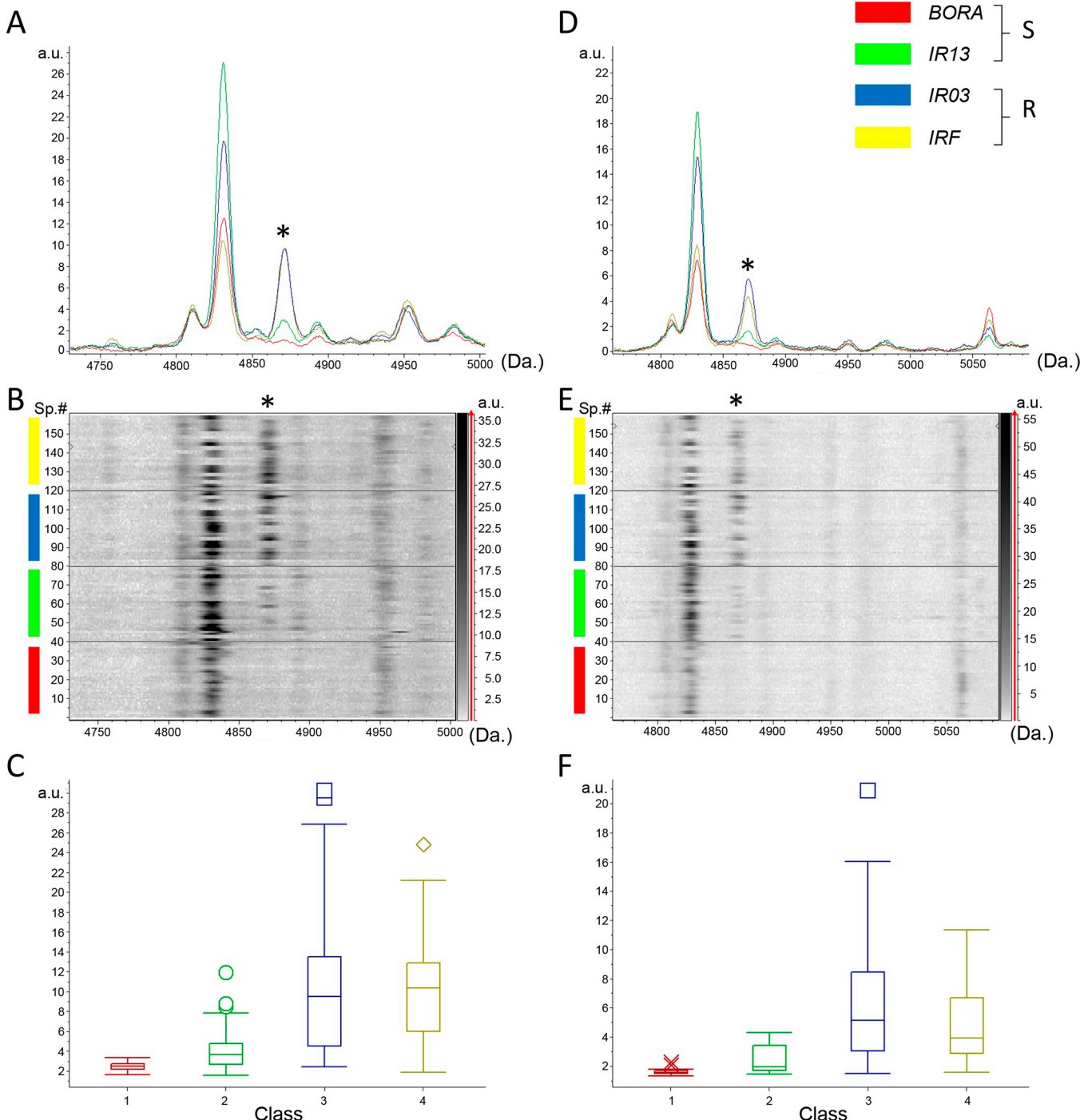

**Fig 3. Variation of the MS peak at about 4870 m/z among the four *Ae. aegypti* lines according to their susceptibility to deltamethrin.** Overlay mean profile view of leg (A) and thorax (D) body parts according to Ae. aegypti lines. Line color code of each mosquito line is indicated in the top right part. Gel view of leg (B) and thorax (E) MS spectra from the 20 specimens per line. The two replicates loaded on the MS plate for each specimen per body part are presented. Spectra number (Sp.#) is indicated at left and peak intensity is illustrated by a grey scale in arbitrary units. The discriminant MS peak (m/z: 4870 Da) is indicated by an asterisk (*). Graphical representation of the intensity of the 4870 m/z MS peak from legs (C) and thoraxes (F) according to Ae. aegypti lines. Standard deviations of intensities are represented by vertical lines. A.U.: Arbitrary units; m/z: Mass to charge ratio; R: Lines classified deltamethrin-resistant; S: Lines classified deltamethrin-susceptible. The same color code was used for all the panels.

## Assessment of blind test strategy to discriminate deltamethrin-resistant from -susceptible lines

MS spectra from two specimens per line and per body part were selected for creation of reference MS spectra (Additional file 1). These MS spectra were selected in order that those from the deltamethrin-resistant lines (IR03 and IRF) possessed the most discriminant peak detected in both body parts at about 4870 m/z, whereas, this MS peak was absent from the two susceptible lines (BORA, IR13). The remaining MS spectra from legs (n = 144, 18 samples per lines (x4) loaded in duplicate (x2)) and from thoraxes (n = 144) were queried against these reference MS spectra. Overall, 98.9% (n = 285/288) of the MS spectra queried against the database, obtained LSVs over 2.0, and all (100%) reached the threshold established for relevant identification (LSVs>1.8) (S1 Fig). The assessment of concordance of classification results (resistant or susceptible) between blind tests and lines revealed an agreement of 77.1% with a Cohen's κ coefficient of 0.542 corresponding to a moderate agreement of the data for legs. Similarly, a moderate agreement (76.4% with a Cohen's κ coefficient of 0.528) was obtained for thoraxes. The sensitivity and specificity of blind test strategy were, respectively, 72.2% and 81.9% for legs and 72.2% and 80.6% for thoraxes using the four Ae. aegypti lines as reference (Table 3).

## Potential association between the 4870 m/z MS peak and kdr mutations

For each specimen tested in the present study, the V410L, V1016G/I, and F1534C kdr mutations were genotyped in an attempt to identify their potential association with the 4870 m/z MS resistance discriminating peak. These SNPs were selected because they are known to be strongly associated to pyrethroid resistance in Ae. aegypti [41–43]. Among the 20 specimens tested for each line, genotyping failed for 5 individuals (i.e. one IR13, one IRF and three IR03). All genotyped specimens from BORA, IR13 and IRF lines were free of kdr mutations in all three sites of VGSC gene (i.e. genotype frequency of 100% for VV/VV/FF haplotype), while 65% of IR03 lines were heterozygotes (haplotype VV/VV/FC) and 6% were homozygote resistant (haplotype VV/VV/CC) for the F1534C mutation (Table 4).

As the four lines were confirmed to be susceptible for the V410L and V1016G/I mutations, the potential association between the F1534C mutation and the 4870 m/z MS peak was investigated. The classification of the leg and thorax spectra according to genotypes (FF, FC or CC) did not show any association between F1534C genotypes and the abundance of 4870 m/z MS

**Table 3. Comparison of the classification of *Ae. aegypti* lines according to deltamethrin-susceptibility.**

| | | Legs | | Thoraxes | |
|---|---|---|---|---|---|
| | | Lines | | Lines | |
| | | Resistant (IR03, IRF) | Susceptible (BORA, IR13) | Resistant (IR03, IRF) | Susceptible (BORA, IR13) |
| Blind tests* | R | 59 | 20 | 58 | 20 |
| | S | 13 | 52 | 14 | 52 |
| Total | | 72 | 72 | 72 | 72 |
| Agreement (%) | | 77.1% | | 76.4% | |
| Cohen's κ # | | 0.542 (Moderate agreement) | | 0.528 (Moderate agreement) | |
| Sensitivity (%) | | 72.2% | | 72.2% | |
| Specificity (%) | | 81.9% | | 80.6% | |

*Results of spectra classification queried against the reference MS spectra included in the DB

#Coefficient of agreement, the agreement level is indicated into brackets, as previously defined [38]. DB, database; MS, mass spectrometry; Resistant, deltamethrin-resistant lines; Susceptible, deltamethrin-susceptible lines.

**Table 4. Genotyping of V410L, V1016I and F1534C *kdr* mutations in the four lines of *Aedes aegypti*.**

| *Ae. aegypti* lines | Genotypes (410/1016/1534) | | | |
|---|---|---|---|---|
| | VV/VV/FF | VV/VV/FC | VV/VV/CC | Others combinations |
| BORA | 20 | 0 | 0 | 0 |
| IR13 | 19 | 0 | 0 | 0 |
| IR03 | 5 | 11 | 1 | 0 |
| IRF | 19 | 0 | 0 | 0 |
| **Total** | **63** | **11** | **1** | **0** |

peak (S2 Fig). Indeed, the 4870 m/z MS peak was detected in half of the individuals carrying the mutation, either in heterozygosis or homozygosis mutant (FC or CC), and in all susceptible genotypes (FF) for both body parts. Altogether, this indicates that the 4870 m/z MS discriminant MS peak is not related to the V410L, V1016G/I, and F1534C kdr mutations.

## Discussion

The success of MALDI-TOF MS profiling for mosquito species identification [44,45], detection of parasitic agents [34,46] and/or for the determination of blood feeding origin of engorged specimens [23,47] led us to investigate the potential of this tool for the detection of insecticide resistance. Here we focused on deltamethrin, the most widely used pyrethroid insecticide for the control of the main arboviral vector, Ae. aegypti. To reduce the impact of the genetic and environmental conditions on the outcomes, we selected three mosquito lines collected in the same region (French Guiana) and having close genetic backgrounds, including one line being susceptible to deltamethrin (IR13) and two confirmed deltamethrin-resistant lines (IR03 and IRF) [29]. In addition, the susceptible laboratory line Bora-Bora (BORA) carrying no resistance allele was tested to generate reference MS profile.

Prior to research MS peak markers associated to insecticide resistance, a quality control of the MS spectra was carried out by evaluating the accuracy of sample identification against a home-made reference spectrum DB. For the first query, no MS spectra from the three French Guiana Ae. aegypti lines were included in the spectra DB as reference. Nevertheless, a correct and relevant identification (LSV ≥1.8) was obtained for 98.8% (158/160) of the samples. The matching of these query leg and thorax MS spectra with those of Ae. aegypti from the DB coming from the laboratory (i.e., BORA) or the field [45] underlined that the spectra were conserved among these Ae. aegypti specimens from distinct origins (reproducibility of spectra). The presence of reference spectra from specimens of the same species was then sufficient for correct and reliable identification of species identification, confirming the compliance of the MS spectra dataset from legs and thoraxes for next analyses [24,27].

The significant higher LSVs obtained for thoraxes compared to legs confirmed the better MS spectra reproducibility of the thoraxes for species identification, as recently demonstrated [26]. Nevertheless, for both body parts, comparisons of CCI, cluster analysis and PCAs indicated that MS spectra were more reproducible among strains coming from the same geographical origin (the three lines from French Guiana) than from BORA line, which is originating from French Polynesia and has been maintained in the laboratory for more than 30 years. Indeed, the MS spectra from the BORA line slightly differed from those of French Guiana (IR03, IR13 and IRF) for both body parts, legs and thoraxes. These data also suggest that MS profiles from isofemale lines are closer among them than between specimens presenting the same properties regarding the deltamethrin susceptibility phenotype. These results are concordant with previous works which already reported that MS spectra from specimens of the same

species were more homogeneous if they have the same geographical origin [21,26]. Then, the higher MS spectra homology according to specimen geographical origin underlined that the classification of specimens according to their deltamethrin susceptibility could not be elucidated by the analysis of the whole MS spectra, as it is commonly done for arthropod species classification by MALDI-TOF MS biotyping [48]. Then, we focused our work by looking at specific MS peaks that could distinguish specimens according to their deltamethrin susceptibility.

The comparison of the average spectrum intensity between the IR03 and IRF deltamethrin-resistant and the BORA and IR13 susceptible lines revealed multiple MS peaks with significant abundance variations for legs and thoraxes. These peaks allowed to classify correctly more than 90.0% of the specimens after applying a GA model on both body parts. These data underlined that less than a dozen of MS peaks per body part appeared sufficient to segregate both groups. Such imperfect classification is probably explained by the inter-individual variability exists in the lines tested and by other resistant markers (e.g. metabolic) which may be present in the mosquitoes originated from French Guiana lines [29].

In this kind of study, the difficulty is coming from the heterogeneity or spectra variation which occurred among specimens from the same species but coming from distinct origins. That's why the main factors that can create spectra noise were controlled by using Ae. aegypti lines having the same genetic background (i.e. isofemales lines) and coming from the same geographical region. Moreover, the laboratory rearing of the four mosquito lines in standardized breeding conditions participated also to reduce spectra variation due to environmental factors. A recent study demonstrated that the application of machine learning models to MS spectra from legs or thoraxes from anopheline mosquitoes could detected biomarkers associated to diverse life history traits such as, the population age, past blood feeding or plasmodium infectious status [49].

Among the discriminant MS peaks, one peak, at about 4870 m/z, was found significantly more intense (fold change upper than 2.8) in deltamethrin-resistant lines as compared to the susceptible one's. This peak was found in both legs and thoraxes. The application of GA model to this peak allowed to classify mosquito specimens as deltamethrin-susceptible or deltamethrin-resistant with a concordance of about 80%. Although promising, we have no guarantee that this peak is systematically and functionally associated to deltamethrin resistance. First of all, no association was found between kdr alleles at position 1534 (at heterozygote or homozygote haplotypes) and the abundance of 4870 m/z MS peak. No mutant kdr alleles were find at 1016 and 410 sites in all isofemale lines from French Guiana as initially reported [29], then no genotype-phenotype association studies were performed. Our data suggest that the deltamethrin resistance phenotype of the IR03 and IRF lines is not majorly caused by kdr mutations but most probably by other metabolic resistance alleles as previously reported [32]. In this regard, it is possible that the abundance of the 4870 m/z MS peak in deltamethrin-resistant lines rather reflects a change in the expression of particular detoxifying enzymes involved in deltamethrin-resistance, such as P450s. This peptide could also result from an indirect effect and could have a priori no causal link with the insecticide resistance phenotype. Clearly, further work is needed to assess the protein nature and function associated with this 4870 m/z MS peak. One way to do this may involve further association studies using the IRF line deprived from kdr mutations together with cross-comparison of MS data with molecular data obtained from loci associated with resistance. The tracking of this particular MS peak in a larger collection of resistant and susceptible populations may also help us to validate this association of this MS peak with resistance.

Finally, the functional characterization of the 4870 m/z MS peak is a priority. Although the identification of the other discriminant peaks could be helpful to establish their involvement

in phenotypic deltamethrin-resistant trait, their lower peak intensity could compromise these characterizations. Conversely, the peak at m/z 4870 was among the top one for legs and top two for thoraxes of discriminant average peak intensity. Moreover, taking into account the whole spectra, this peak was among the more abundant. The identification of the protein/peptide will however require additional MS apparatus [50]. For example, the protein/peptide could identify by peptide-sequencing using tandem mass-spectrometry approach [51] and validated by immune detection (eg, ELISA, WB, etc) [52]. Regardless the technique, the identification and incrimination of this peak in deltamethrin-resistance could open the door for the development of novel diagnostic assays to track pyrethoid resistance.

## Conclusions

The MALDI-TOF MS profiling is an innovative approach which proved to be a rapid, affordable and efficient method for the identification of vector species, blood feeding source and some pathogen infection. This emerging entomological strategy may also be relevant for identifying others mosquito life traits of major importance, such as insecticide resistance. Although, the analysis of global MS profiles failed to distinct susceptible to resistant phenotypes in the inbreed females, an accurate comparison of spectra allowed to reveal potential peaks associated to deltamethrin-resistance. This pioneering study requires further complementary experimental works and collaborative research efforts, to consider the potential outputs for the mosquito surveillance. The characterization of key mosquito life traits with a unique approach will be revolutionary for vector biology and the prevention of mosquito-borne diseases outbreaks.

## Supporting information

**S1 Fig.** LSVs following upgrading homemade reference database with MS spectra from legs (A) and thoraxes (B) of the four *Ae. aegypti* lines. Eighteen specimens per line were tested. Horizontal dashed lines represent the threshold value for reliable identification (LSV>1.8). Red, green, blue and yellow dots correspond to BORA, IR13, IR03 and IRF *Ae. aegypti* lines, respectively. LSVs, log score values; a.u., arbitrary units.
(PPTX)

**S2 Fig. Variation of the MS peak at about 4870 m/z among the IR03 *Ae. aegypti* line according to their 1534 *kdr* genotyping.** Overlay mean profile view of leg (A) and thorax (D) body parts according to 1534 genotype. Line color code of each genotype is indicated in the top right part. Gel view of leg (B) and thorax (E) MS spectra from the IR03 specimens per genotype. The two replicates loaded on the MS plate for each specimen per body part are presented. The discriminant MS peak (m/z: 4870 Da) is indicated by an asterisk (*). Graphical representation of the intensity of the 4870 m/z MS peak from legs (C) and thoraxes (F) according to 1534 genotype of IR03 *Ae. aegypti* line. Standard deviations of intensities are represented by vertical lines. A.U.: arbitrary units; m/z: mass to charge ratio; Red: haplotype homozygotes without mutation (VV/VV/FF); Green: mutant haplotype heterozygotes (VV/VV/FC) and homozygotes (VV/VV/CC). The same color code was used for all the panels.
(PPTX)

**S1 Table. Primer pairs used for detection of *kdr* mutation points.**
(DOCX)

**S1 File. Raw leg and thorax MS spectra from the four *Ae. aegypti* lines included in the MS reference database.** MS spectra were obtained using Microflex LT MALDI-TOF Mass Spectrometer (Bruker Daltonics). Details of each sample were listed of the excel file named

"REF_MS_Spectra_Mosq_Guiana_Body-parts_IRS_August-2023".
(ZIP)

## Acknowledgments

We thank the "Fondation Méditerranée Infection (FMI)" and the WIN (Worldwide Insecticide resistance Network) for their administrative support.

## Author Contributions

**Conceptualization:** Lionel Almeras.

**Data curation:** Lionel Almeras, Rémy Amalvict, Jean-Philippe David, Vincent Corbel.

**Formal analysis:** Lionel Almeras, Monique Melo Costa, Rémy Amalvict, Joseph Guilliet, Isabelle Dusfour, Jean-Philippe David, Vincent Corbel.

**Funding acquisition:** Lionel Almeras.

**Investigation:** Lionel Almeras, Isabelle Dusfour.

**Methodology:** Lionel Almeras, Rémy Amalvict, Joseph Guilliet, Jean-Philippe David, Vincent Corbel.

**Project administration:** Lionel Almeras.

**Resources:** Lionel Almeras, Joseph Guilliet, Isabelle Dusfour, Jean-Philippe David, Vincent Corbel.

**Supervision:** Lionel Almeras, Vincent Corbel.

**Validation:** Lionel Almeras, Jean-Philippe David, Vincent Corbel.

**Visualization:** Lionel Almeras.

**Writing – original draft:** Lionel Almeras, Vincent Corbel.

**Writing – review & editing:** Monique Melo Costa, Joseph Guilliet, Isabelle Dusfour, Jean-Philippe David, Vincent Corbel.

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
