## [Decision Letter · Decision Letter 0]

22 Mar 2024

PONE-D-24-05959Potential of MALDI-TOF MS biotyping to detect deltamethrin resistance in the dengue vector Aedes aegypti.PLOS ONE

Dear Dr. ALMERAS,

Thank you for submitting your manuscript to PLOS ONE. After careful consideration, we feel that it has merit but does not fully meet PLOS ONE’s publication criteria as it currently stands. Therefore, we invite you to submit a revised version of the manuscript that addresses the points raised during the review process.

We look forward to receiving your revised manuscript.

Kind regards,

Joseph Banoub, Ph,D., D. Sc.

Academic Editor

PLOS ONE

Journal Requirements:

"AL received the following award

This work has been supported by the Délégation Générale pour l’Armement (DGA), MSProfileR project, Grant no PDH-2-NRBC-2-B-2201

This work was also supported by the WIN (Worldwide Insecticide resistance Network)"

"We thank the “Fondation Méditerranée Infection (FMI)” which offered personnel grant to MMC and the WIN (Worldwide Insecticide resistance Network) which contributed in the travelling of the student between Brazil and France."

Please remove any funding-related text from the manuscript.

5. We are unable to open your Supporting Information file [Additional_file_1_DB.7z]. Please kindly revise as necessary and re-upload.

**Additional Editor Comments:**

It will be appreciated to revise your manuscript according to all suggestion of the referees.

Reviewers' comments:

Reviewer's Responses to Questions

**Comments to the Author**

1. Is the manuscript technically sound, and do the data support the conclusions?

Reviewer #1: Partly

Reviewer #2: Partly

2. Has the statistical analysis been performed appropriately and rigorously? 

Reviewer #1: Yes

Reviewer #2: I Don't Know

3. Have the authors made all data underlying the findings in their manuscript fully available?

Reviewer #1: Yes

Reviewer #2: Yes

4. Is the manuscript presented in an intelligible fashion and written in standard English?

Reviewer #1: Yes

Reviewer #2: Yes

5. Review Comments to the Author

Reviewer #1: The article "Potential of MALDI-TOF MS biotyping to detect deltamethrin resistance in the dengue vector Aedes aegypti" describes a new diagnostic approach based on protein profile signatures using MALDI-TOF-MS instrument and identification by an MS reference spectra database to determine pyrethroid insecticide resistance in mosquitoes. In this context, the study suggests performing a comparison of various MS profiles from four Ae. aegypti colonies, backed by statistical analysis tools. This audacious proposal seeks an alternative advanced diagnostic method using protein profiling data analysis by MALDI-TOF-MS to detect resistant mosquitoes at high-throughput levels.

The scientific work is really promising and crucial. Overall, the text is well written and abundant in experiment and outcome data. However, I believe the text is incomplete; certain decisive conclusions are not effectively interpreted and require more, pertinent interpretation. I agree to have this work published in the journal PLOS ONE only if the following adjustments are made and the issues are addressed.

Introduction, pages 3 to 4:

Line 53: missing a reference

Line 59: “…in vector control for decades has selected mosquito…” do you mean developed

Line 96: replace deltamethrin with pyrethroid ester insecticide

Sample homogenization and MALDI-TOF MS, pages 5-6 lines 137-150:

How did you perform the MALDI-TOF-MS calibration during the experiment? Please add the calibration method to the paper.

Database creation and blind tests pages 6 to 7 lines 173-183:

-The quality of MALDI-TOF mass spectra may be affected by the following factors: technical knowledge of acquiring MALDI-TOF mass spectra, such as regular staff training and quality control of MALDI-TOF MS measurements, factors affecting spectrum acquisition settings, such as the number of laser shots applied and spectra averaged per measurement. Have you carried out routine diagnostics (QC tests) using standardized MALDI-TOF MS spectral quality?

-Did you prepare any pooled sample by homogenizing both parts legs and thorax? This pooled sample could be applied as a quality control for MS data correction and normalization.

Results, pages 8-11 lines 195-296:

-Lines 222-224: “These results highlighted that overall comparisons of MS profiles did not allow to clearly distinguishing specimens according to their deltamethrin resistance status whatever the body- part tested.” I do not agree with the statement above. In my opinion, your spectra will at least show the peak height intensity difference if you normalize them.

-According to the PCA, there is a significant difference between BORA (French Polynesia) and the three others (French Guiana). I think you have multivariable factors to be taken into consideration.

-Have you tried multivariable study?

-Have you tried Partial Least-Squares Discriminant Analysis (PLS-DA)?

-Line 263: “m/z of about 4870 Da”, I see in Table 1 that peak #16 is at m/z 4871.1 (TOF analyzer associated in general with high-resolution MS)

-Please add peak annotation (at least peptide sequence)

-Lines 253-257, Table 2 peak #29, I see m/z 4869.5 and not m/z 4870.

In my opinion, peaks #16 and #29 observed in both leg and thorax might be different

-Page 11, lines 278-296 Potential association between the 4870 m/z MS peak and kdr mutations: Indeed, this is a very important study; did you consider associating other m/z significant MS peaks? Why only 4870 m/z? the other m/z values have a fold change of more than one.

-The discriminant MS peaks (observed in Tables 1&2 form legs and thorax deltamethrin-resistant (IR03 and IRF) and susceptible (BORA and IR13) Ae. aegypti line (BORA and IR13) Ae. aegypti lines) are all significant and they all present a potential biomarker for the resistant-group species. In this case, you should reconsider the data interpretation to value more these findings.

Discussion

Line 361, the discussion should be extended to other m/z ions observed in table 1 and 2

Conclusion: the conclusion is very general and brief

-Rewrite the conclusion to highlight the importance of MALDI-TOF-MS in monitoring deltamethrin resistance in the dengue vector Aedes aegypti. I agree that this pioneering effort requires further development in the future. However, you should put more value to the findings in this work especially related to the m/z list found in Tables 1 and 2, which in my opinion should be more discussed and analyzed pertinently. This work is missing a biological survey of the other peak discovered, which would provide us with a better understanding of the true biomarker associated with deltamethrin resistance.

Figures

-Figure 1: The mass spectra have different intensity scales; please consider normalizing the MS by putting a common intensity scale for better comparison.

-Figure 2: PCA is not clear and the legend is very small, please reconsider the font size of the figure to be better readable.

-Figure 3 (A &D) is not clear, it is difficult to read the m/z for the peak you mentioned.

Reviewer #2: In this paper, the authors have utilised MALDI-TOF-MS protein/peptide fingerprinting in an attempt to find any discriminating peaks between different Ae. aegypti deltamethrin resistant and susceptible lines. The analysis was performed using two body parts, thorax and legs. The resulting spectra was compared and analysed using various statistical tools leading to the identification of a peak at m/z 4870 representing the higher fold change (>2.8 fold) between resistant and susceptible-lines for both body parts.

I have some points that need to be clarified regarding the MS data analysis and the two peaks at m/z 4871.1 and m/z 4869.5 detected from legs and thorax, respectively.

1- The difference between m/z 4871.1 and m/z 4869.5 is 1.6 Da. This indicates that these two peaks could be completely different as most peptide searches are usually done within +/-0.2 Da. So, what are the criteria used to decide if two peaks are similar or not?

Also, it is not clear in the manuscript what mass tolerance used for searching the spectra against the reference MS spectra from the 16 distinct mosquito species. Is this database available for everyone to use and approved for reliable species identification? Is it just a list of m/z values? Does it include Protein/Peptide IDs? Were the IDs supported by tandem mass spectrometry (MS/MS)?

2- These two peaks (m/z 4871.1 and m/z 4869.5) have extremely low signal intensities in the range of 2-11 a.u (Tables 1 and 2). So, what are the parameters used to evaluate the data and/or to filter real signals from noise? i.e. what is the minimum S/N ratio, signal intensity or relative intensity to create a peak list of 99 and 118 peaks detected in legs and thoraxes, respectively?

6. PLOS authors have the option to publish the peer review history of their article (what does this mean?). If published, this will include your full peer review and any attached files.

Reviewer #1: No

Reviewer #2: No

---

## [Author Response · Author response to Decision Letter 0]

15 Apr 2024

To support our responses, new figures were added in the responses to reviewer comments, then the responses to reviewer comments and journal asks were included in a response letter loaded at the "attach files" step with the new version of the manuscript. The file name was headed “Responses_to_reviewer Comments_R1_PONE”.

---

## [Editor Report · Decision Letter 1]

18 Apr 2024

Potential of MALDI-TOF MS biotyping to detect deltamethrin resistance in the dengue vector Aedes aegypti.

PONE-D-24-05959R1

Dear Dr. ALMERAS,

We’re pleased to inform you that your manuscript has been judged scientifically suitable for publication and will be formally accepted for publication once it meets all outstanding technical requirements.

I really would like to thank you for the excellent answers to the queries of the referees of this manuscript. It was quite informative and very well written.

Kind regards,

Joseph Banoub, Ph,D., D. Sc.

Academic Editor

PLOS ONE
---

## [Editor Report · Acceptance letter]

29 Apr 2024

PONE-D-24-05959R1 

PLOS ONE

Dear Dr. ALMERAS, 

I'm pleased to inform you that your manuscript has been deemed suitable for publication in PLOS ONE. Congratulations! Your manuscript is now being handed over to our production team.

Kind regards, 

on behalf of

Dr. Joseph Banoub 

Academic Editor

PLOS ONE